

# Adequacy of clinical information in X-ray referrals for traumatic ankle injury with reference to the Ottawa Ankle Rules—a retrospective clinical audit

Yolanda E. Gomes[1], Minh Chau[1], Helen A. Banwell[1], Josephine Davies[2] and Ryan S. Causby[1]

[1] Allied Health and Human Performance Unit, University of South Australia, Adelaide, South Australia, Australia
[2] South Australia Medical Imaging, Flinders Medical Centre, Bedford Park, South Australia, Australia

Corresponding author
Ryan S. Causby,
Ryan.Causby@unisa.edu.au

## ABSTRACT

**Study Objective**. To assess the adequacy of clinical information with reference to the Ottawa Ankle Rules (OAR) in X-ray referrals for adults with traumatic ankle injury in the ED of a South Australian tertiary hospital and report upon referring trends between emergency department clinicians.

**Methods**. A retrospective clinical audit of adult ankle X-ray referrals in the emergency department was conducted. Eligible referrals were screened for their adherence to the OAR, patient details, clinical history and referrer. A logistic regression was used to determine the influence of these factors on the likelihood of being referred for X-rays despite not meeting the OAR criteria. Sensitivity, specificity, positive and negative likelihood ratios and their associated confidence intervals were calculated to assess the diagnostic accuracy of the OAR for those referred.

**Results**. Out of the 262 eligible referrals, 163 were deemed to have met the criteria for the OAR. Physiotherapists showed the highest OAR compliance of 77.3% and were the most accurate in their use of the rules, with a sensitivity of 0.86. Medical officers, registrars and interns were 2.5 times more likely to still refer a patient for X-ray if they did not meet the OAR criteria, compared to physiotherapists as the baseline. Patient age, duration of injury etc. were not significantly associated with likelihood of referral (even when they did not meet OAR criteria). The overall sensitivity, specificity, positive and negative likelihood ratios of the OAR were 0.59 (95% CI [0.47–0.71]), 0.37 (95% CI [0.30–0.44]), 0.93 (95% CI [0.76–1.16]) and 1.10 (95% CI [0.82–1.48]) respectively.

**Conclusion**. The results of this audit demonstrated poor sensitivity and moderate compliance by referrers with the rule. Reasonable evidence exists for the implementation of individual and/or institutional-based change strategies to improve clinician compliance and accuracy with use of the OAR.

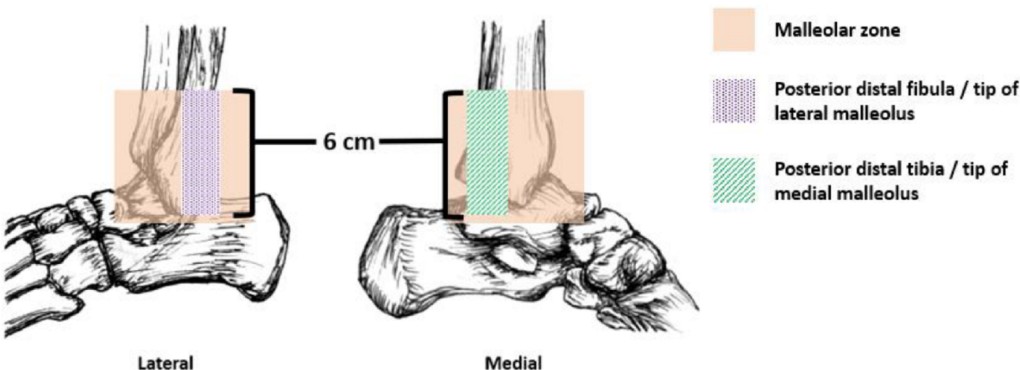

**Figure 1 The Ottowa Ankle Rules.** The Ottowa Ankle Rules state that ankle radiographs are required if the patient experiences malleolar pain and bone tenderness at the posterior distal fibula/tip of lateral malleolus, posterior distal tibia/tip of medial malleolus or an inability to weight bear for four steps both immediately and in the emergency department.

# BACKGROUND

Ankle and foot injuries are the most frequently presenting musculoskeletal injuries in Australian emergency departments (ED) (*Strudwick et al., 2018*). Over 46,000 presentations occurred in South Australia in the year 2017–18 (*Australian Institute of Health and Welfare, 2018*). Acute ankle trauma is often a result of inversion injury, commonly causing a sprain or disruption of ligaments. Ankle fractures, however, are more likely observed with blunt ankle trauma, such as those associated with sporting injuries or motor vehicle related accidents (*Goergen et al., 2015*). Acute ankle injuries are commonly diagnosed by subjective patient history, objective physical examination and/or using radiographic imaging (*Goergen et al., 2015*).

Radiographic imaging is one of the most routinely used assessment methods for ankle trauma (*Beckenkamp et al., 2017*). Despite the benefits of this modality, the continuous referral of patients for imaging of ankle trauma often leads to increased waiting times in the ED already been defined in the first sentence of background above, contributes to rising healthcare costs and unnecessarily exposes patients to ionising radiation (*Beckenkamp et al., 2017*). Consequently, it is imperative that the management of ankle trauma within EDs is optimised to facilitate best management, minimise costs and improve the quality of care provided to patients (*Strudwick et al., 2018*).

The Ottawa Ankle Rules (OAR) (Fig. 1) are part of a clinical decision-making tool to help clinicians accurately rule out ankle fractures, potentially precluding the need for diagnostic X-ray imaging (*Bachmann et al., 2003*). The OAR were introduced by *Stiell et al. (1992)* and have proven to be a highly accurate tool, with good sensitivity and demonstrated ability to reduce imaging requests and waiting times in the ED (*Cheng, Varma & Smith, 2016*; *Strudwick et al., 2018*).

The OAR have been widely applied in many countries (*Dap, Temiz & Çevik, 2016*) and there have been active dissemination and education strategies developed to encourage clinicians to incorporate them into practice (*Cameron & Naylor, 1999*). However, *Cameron*

*& Naylor (1999)* report that OAR have not been universally adopted into practice, potentially due to the convenience of referring patients with ankle trauma for imaging, or due to practitioners' concerns around litigation (*Pires et al., 2014*). Evidence suggests that in order to influence changes in clinical behaviour, implementing guidelines on a local level is paramount (*Cameron & Naylor, 1999*).

A cross-sectional study evaluated the international adoption of the OAR and reported significant differences in the use of the rules by physicians in five countries. More than 80% of physicians in the United Kingdom and Canada that were aware of the OAR, reported using them frequently. However, the usage was far lower for physicians in the United States (31%), France (31%) and Spain (9%) (*Graham et al., 2001*).

Evidence of uptake in the Australian context is limited. An Adelaide-based validation study found the rules were correctly applied by both junior and senior ED physicians; 47 of 54 physicians (87%) correctly interpreted the requirement for radiography in 327 patients (97.3%) (*Broomhead & Stuart, 2003*). In a separate study, analysis of the use of the OAR by nurse practitioners, triage nurses and other medical staff identified a gap between evidence and practice; with reasons for ordering radiographs including an obligation to the patient, streamlining patient flow through the ED and wanting to avoid patients 're-presenting' with the injury (*Bessen et al., 2009*). Furthermore, a retrospective review of a major metropolitan ED in Victoria showed positive results, with a 87.9% compliance rate with the OAR (*Cheng, Varma & Smith, 2016*).

To the best of the authors' knowledge, however, the adequacy of clinical information on adult ankle radiograph requests with reference to the OAR has not been recently investigated in a South Australian context. Therefore, the aims of this study are to:

(i) assess the current usage of the OAR in ruling out ankle fractures in a major South Australian metropolitan tertiary care emergency department,

(ii) evaluate the current concordance rate of scoring with positive findings on radiography, and

(iii) report upon referring trends between professions, including consultants, registrars/medical officers/interns, physiotherapists and nurse practitioners.

These research findings can provide the first step into further research on the awareness, dissemination and uptake of the OAR in Australia.

## MATERIALS & METHODS

A retrospective clinical audit was performed on X-ray referrals for adult patients presenting with acute ankle injuries to the emergency department of Flinders Medical Centre (a tertiary hospital in South Australia). Approval was granted by the University of South Australia Human Research Ethics Committee (Approval number: 202798) and waived by the Southern Adelaide Local Health Network Human Research Ethics Committee of Flinders Medical Centre. A confidentiality report was signed by the data collector and all patient information was de-identified. The requirement for obtaining informed consent was waived.

Referrals for imaging were excluded from the study if participants were under 18 years of age, returning for follow-up imaging, presenting with a multi-trauma, had an injury

more than two weeks old, had cognitive impairment (including intoxication), had an incomplete examination, had an inflammatory, neurological or musculoskeletal condition that impeded ankle joint function, the referral gave no fracture subscription (i.e., query foreign body, prosthesis position), or requested imaging for a known dislocation or post-reduction of a dislocation. This is consistent with previous literature (*Dap, Temiz & Çevik, 2016*, p. 362; *Pijnenburg et al., 2002*, p. 601).

The hospital's Picture Archiving and Communication System (PACS) was reviewed in January 2020 to extract relevant ankle X-ray referrals from March 2019 to January 2020. The referrals were screened for their adherence to the OAR and patient details. Data extracted included patients' date of birth, clinical history and the type of referring health professional (i.e., consultant, registrar/medical officer (MO)/intern, physiotherapist or nurse). A referral was deemed to have met the criteria for the OAR if it (i) indicated pain in the malleolar zone and (ii) indicated pain or bone tenderness of the posterior distal tibia/medial malleolus tip or the posterior distal fibula/lateral malleolus tip or an inability to weight bear for 4 steps both immediately and in the ED (*Tiemstra, 2012*). All data was entered into Excel (Microsoft Corporation 2016). The radiologist's report was also screened for the presence/absence of an acute fracture.

Descriptive and inferential statistics were evaluated using SPSS software (IBM SPSS Statistics for Windows, Version 26). Descriptive analyses were conducted for patient demographics (i.e., mean age, duration of injury, etc.). Sensitivity, specificity, positive and negative likelihood ratios were calculated to determine the diagnostic accuracy of the OAR (*Lowry, 2004*) on referrals received. A logistic regression was also performed to assess the impact of other potential decision-making factors when referring for X-rays in patients who did not meet the OAR criteria. Outcomes were considered statistically significant if $p < 0.05$.

## RESULTS

The search yielded 750 referrals. Of the 750 referrals screened, 488 were excluded (Fig. 2). The characteristics of the 262 eligible referrals are summarised in Table 1.

Participants' mean age was 38 (SD 13.8) and the ages ranged from 18-97 years. Males and females were equally represented. Over three-quarters of referrals for imaging did not specify the duration of injury and rolling or twisting injuries were the most commonly recorded mechanism of injury (42.4%), with inversion/eversion injuries also frequently reported (32.4%). Registrars, MOs and interns were grouped into one category due to an inability to differentiate between them on referrals. As such, they were the largest referring profession (47.3%), followed by nurses (30.9%).

Out of the 262 eligible referrals for imaging, 163 met the criteria for the OAR (OAR +), while 99 did not (OAR-). This suggests that 38% (99 of 262) referrals for imaging may not have been necessary. Table 2 summarises the characteristics of the two groups.

Registrars, MOs and interns were the most frequently referring profession, while nurses constituted the second-highest reporting profession in both groups. Physiotherapists were the most compliant profession in their use of the OAR criteria (77.3%).

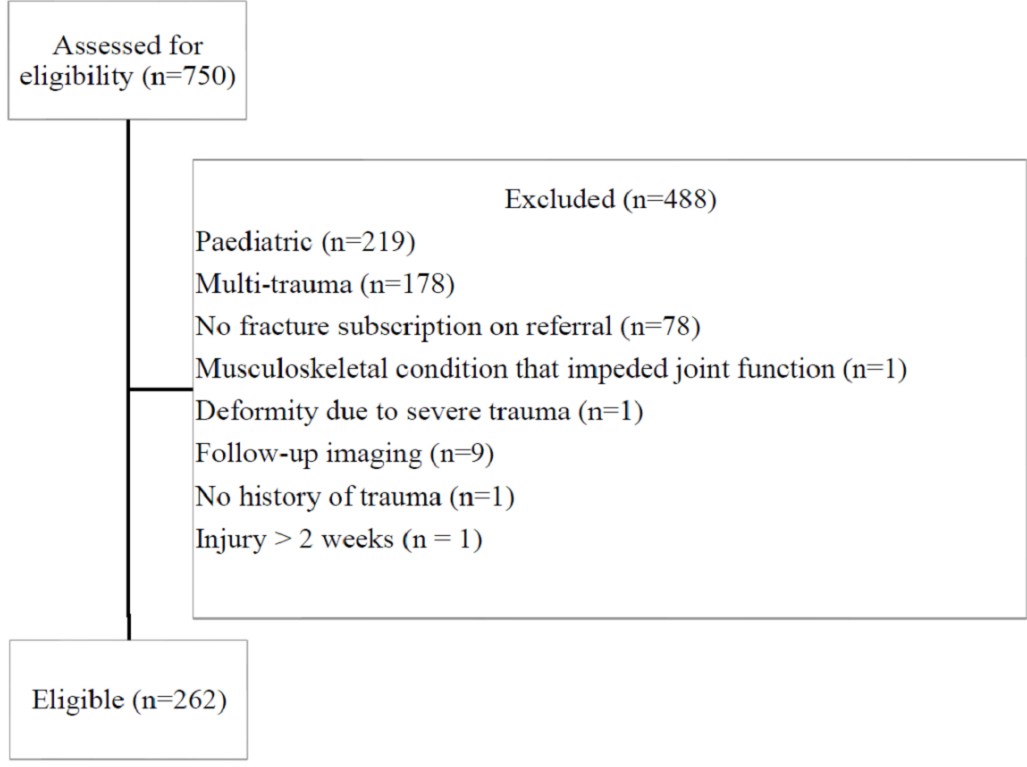

**Figure 2** **Referral selection criteria.** An outline of the reviewed referrals and reasons for exclusion from the study.

Nurses and consultants showed 65.4% and 61.5% compliance respectively, while registrars/MOs/interns showed the least compliance at 54.8%.

A logistic regression was performed to assess the impact of clinical factors on referring for X-rays in patients who did not meet the OAR criteria. The model contained several independent variables (duration of injury, mechanism of injury, referring profession and participant age). The full model containing all predictors was statistically significant, $\chi^2$ (9, $N = 262) = 16.9, p = 0.05$, indicating the model was able to distinguish between OAR+ and OAR- referrals. As shown in Table 3, the only statistically significant independent variable that contributed to the model was registrars, MO's and interns, recording an odds ratio of 2.48 ($p = 0.026$). This indicates that this group of professions was 2.5 × more likely to still refer a patient for X-ray if they did not meet the criteria, compared to physiotherapists as the baseline.

The number of referrals for imaging that resulted in identified ankle fractures are summarised in Table 4.

Of the 163 OAR+ referrals, 44 (27.0%) resulted in a fracture or potential fracture as per the radiologist's report, while 119 (73.0%) reported no fracture. Of the 99 referrals that were OAR-, 30 (30.3%) reported a fracture/potential fracture, while 69 (69.7%) did not report a fracture. Overall, the sensitivity and specificity of the OAR were 0.59 (95% CI [0.47–0.71]) and 0.37 (95% CI [0.30–0.44]) respectively. The positive (LR+) and negative

**Table 1  Referred patient characteristics.** A table outlining numbers and percentage for referral characteristics.

| Characteristics | Number | % |
| --- | --- | --- |
| Mean age in years (SD) | 38 (±18.3) | |
| Gender | | |
| - *Male* | 132 | 50.4 |
| - *Female* | 130 | 49.6 |
| Duration of Injury | | |
| - *None specified* | 200 | 76.3 |
| - *<24 h* | 24 | 9.2 |
| - *1–7 days* | 34 | 13.0 |
| - *7–14 days* | 4 | 1.5 |
| Mechanism of injury (as stated on referral) | | |
| - *Fall* | 40 | 15.3 |
| - *Motor vehicle accident (MVA)* | 3 | 1.1 |
| - *Sporting injury* | 23 | 8.8 |
| - *Inversion/eversion injury* | 85 | 32.4 |
| - *Other (i.e., rolling, twisting)* | 111 | 42.4 |
| Referring profession | | |
| - *Nurse* | 81 | 30.9 |
| - *Consultant* | 13 | 5.0 |
| - *Registrar/MO/Intern* | 124 | 47.3 |
| - *Physiotherapist* | 44 | 16.8 |

(LR-) likelihood ratios were 0.93 (95% CI [0.76–1.16]) and 1.10 (95% CI [0.82–1.48]) respectively.

The referrals were aggregated according to the referring clinician to assess the clinician-specific accuracy of the OAR. The sensitivity, specificity, LR+ and LR- of the OAR when applied by each emergency clinician were calculated along with 95% confidence intervals, as were true positives (TP), false positives (FP), true negatives (TN) and false negatives (FN). This information is displayed in Table 5.

Physiotherapists were the most accurately reporting profession (sensitivity 0.86), followed by consultants (sensitivity 0.75).

## DISCUSSION

The OAR have been validated on an international scale and are regarded as a highly sensitive clinical decision tool with the capacity to reduce the number of unnecessary ankle radiographs ordered (*Beckenkamp et al., 2017*). While the application of the OAR by medical doctors in an emergency setting has been previously validated, *Beckenkamp et al. (2017)* highlight the importance of the accurate application of the rules by triage nurses and physiotherapists, as they play an increasingly important role in the outflow of patients within EDs. The uptake of the OAR and application of the rules by different health professionals has been sparsely validated in the Australian context.

**Table 2  Characteristics of patients meeting or not meeting OAR criteria.** Numbers and percentages of characteristics for referrals meeting or not meeting OAR criteria.

| Characteristics | OAR + | | OAR - | |
|---|---|---|---|---|
| | Number | % | Number | % |
| Mean age (SD) | 39 (±18.3) | | 39 (±18.2) | |
| Gender | | | | |
| - *Male* | 81 | 49.7 | 51 | 51.5 |
| - *Female* | 82 | 50.3 | 48 | 48.5 |
| Duration of Injury | | | | |
| - *None specified* | 118 | 72.4 | 82 | 82.8 |
| - *<24 h* | 21 | 12.9 | 3 | 3.0 |
| - *1–7 days* | 22 | 13.5 | 12 | 12.1 |
| - *7–14 days* | 2 | 1.2 | 2 | 2.0 |
| Mechanism of injury (as stated on referral) | | | | |
| - *Fall* | 22 | 13.5 | 18 | 18.2 |
| - *MVA* | 2 | 1.2 | 1 | 1.0 |
| - *Sporting injury* | 10 | 6.1 | 13 | 13.1 |
| - *Inversion/eversion injury* | 63 | 38.7 | 22 | 22.2 |
| - *Other (i.e., rolling, twisting)* | 66 | 40.5 | 45 | 45.5 |
| Referring profession | | | | |
| - *Nurse* | 53 | 65.4 | 28 | 34.6 |
| - *Consultant* | 8 | 61.5 | 5 | 38.5 |
| - *Registrar/MO/Intern* | 68 | 54.8 | 56 | 45.2 |
| - *Physiotherapist* | 34 | 77.3 | 10 | 22.7 |

**Table 3  Logistic regression for criteria relating to referrals.** An outline of the statistical analysis results providing Odds Ratios, Confidence Intervals and *P*-values for referral characteristics.

| | | Odds Ratio | 95% Confidence Interval | | *P*-value |
|---|---|---|---|---|---|
| | | | Lower | Upper | |
| | Age | 0.998 | 0.983 | 1.014 | 0.825 |
| | Duration of injury | 0.833 | .589 | 1.179 | .304 |
| | Fall | 1.185 | .551 | 2.550 | .663 |
| | MVA | .605 | .047 | 7.806 | .700 |
| Mechanism of injury | Sport-related | 1.817 | .702 | 4.703 | .218 |
| | Inversion/eversion | 0.532 | .277 | 1.018 | .057 |
| | Other | 1.00 | . | . | . |
| | Nurse | 1.809 | .762 | 4.296 | .179 |
| Referring profession | Consultant | 2.099 | .519 | 8.486 | .298 |
| | Registrar/MO/Intern | 2.484 | 1.113 | 5.544 | **.026**[*] |
| | Physiotherapist | 1.00 | . | . | . |

**Notes.**
[*] <0.05.
[**] <0.001.

Gomes et al. (2020), *PeerJ*, DOI 10.7717/peerj.10152        7/12

**Table 4  Summary of OAR criteria on referrals and prevalence of clinically significant fractures.** Table of referrals meeting or not meeting OAR criteria cross-tabulated with positive and negative fracture findings.

| OAR criteria on X-ray referrals | Fracture | No fracture | Total |
|---|---|---|---|
| OAR + | 44 | 119 | 163 |
| OAR - | 30 | 69 | 99 |
| Total | 74 | 188 | **262** |

**Table 5  Diagnostic accuracy of OAR when applied by different health professionals.** A table of diagnostic accuracy for the OAR criteria, by health professional referrals, including Sensitivity, Specificity and Likelihood Ratios.

|  | Nurse | Consultant | Registrar/MO/Intern | Physiotherapist |
|---|---|---|---|---|
| No. of patients | 81 | 13 | 124 | 44 |
| TP | 13 | 3 | 16 | 12 |
| FP | 40 | 5 | 52 | 22 |
| TN | 22 | 4 | 35 | 8 |
| FN | 6 | 1 | 21 | 2 |
| Sensitivity | 0.68 (0.43–0.86) | 0.75 (0.22–0.99) | 0.43 (0.28–0.60) | 0.86 (0.56–0.97) |
| Specificity | 0.35 (0.24–0.49) | 0.44 (0.15–0.77) | 0.40 (0.30–0.51) | 0.27 (0.13–0.46) |
| LR+ | 1.06 (0.74–1.52) | 1.35 (0.60–3.04) | 0.72 (0.48–1.09) | 1.17 (0.86–1.58) |
| LR- | 0.89 (0.43–1.82) | 0.56 (0.078–4.03) | 1.41 (1.02–1.93) | 0.54 (0.12–2.31) |

**Notes.**

TP, True Positive; FP, False Positive; TN, True Negative; FN, False Negative; LR+, Positive Likelihood Ratio; LR-, Negative Likelihood Ratio.

This clinical audit assessed the current use of the OAR in a major South Australian tertiary hospital. Based on our findings, physiotherapists showed the highest OAR compliance of 77.3% ($n = 34$) and were the most accurate in their use of the rules, with the highest sensitivity of 0.86. Consultants, although not the most compliant profession (61.5%), displayed a reasonably high sensitivity of 0.75. They also referred the least number of patients ($n = 13$), while registrars/MOs/interns referred the largest proportion of patients ($n = 124$). We hypothesise that this may be due to the distribution of highly complex cases (e.g., motor vehicle accidents) to consultants, while junior doctors are responsible for the triage of a wider variety of less complex cases (e.g., rolling, twisting injuries). This may have biased our results. Nurses and registrars/MOs/interns were the least accurate in their use of the rules, with a sensitivity of 0.68 and 0.43 respectively, which may be due to the inexperience of junior physicians interpreting and applying the OAR.

The most recent prospective validation of the OAR in a South Australian ED, conducted by *Broomhead & Stuart (2003)*, reported a sensitivity of 100% (95% CI [77–100]) and a specificity of 15.8% (95% CI [11–21]) for ankle fractures. Given our investigation was a retrospective and not a prospective study, the sensitivity and specificity of the rules could not be calculated based on the follow-up of participants that did not receive X-rays. A bias may exist among referrals that were deemed non-compliant (i.e., did not meet the criteria for the OAR), as there is the possibility that clinicians used the OAR during triage and whilst
X-rays were not indicated still made a request based on other clinical reasoning concerns. Previous research suggests that an inability to weight-bear is the most important factor influencing referral (*Pires et al., 2014*). We found that 'bone tenderness at the posterior distal fibula or tip of lateral malleolus' was the most common OAR item reported across referrals (44.1%), however, we did not analyse the factors independently.

Despite the large numbers of referrals deemed not compliant to the OAR, our findings did not suggest other possible clinical indicators such as patient age, nor duration or mechanism of injury affected this. The only influencing factor for referral when independent of OAR as an indicator was the referring profession. Results of the logistic regression found that medical officers, registrars and interns were 2.5 times more likely to still refer a patient for X-ray if they did not meet the OAR criteria, compared to physiotherapists as the baseline. As discussed by *Pires et al. (2014)*, this may be due to the convenience of requesting imaging for ankle trauma and/or the fear of litigation. In these instances, the indicator for X-rays is unclear and decreases the accuracy with which we can report on the sensitivity and specificity of the rules. A retrospective study, however, decreases the influence of the Hawthorne effect around application of the OAR. The influence of subjective examiner perception on the referral for X-rays should also be noted. For example, the subjective examination of pain can vary between examiners and differences in clinical skills and experience may impact the perception of fracture occurrence (*Pires et al., 2014*). It is therefore likely that differences in examiner perception during the clinical examination influenced the referral for X-rays.

Within the limits of this study, this audit provides a good summary of the use of the OAR by different emergency clinicians. It provides a starting point for potential further study into the reasons for or against the use of the OAR, particularly amongst registrars/MO's/interns, as well as the diagnostic accuracy of the OAR within a South Australian context. Our study did not involve teaching the correct use and interpretation of the OAR and hence, solely evaluated clinicians' individual ability to correctly apply the rules at baseline to patients presenting with ankle trauma. To improve accuracy and compliance rates, a thorough investigation into the current knowledge and application of the OAR is recommended. Prospective validation of the OAR could follow a small-scale change strategy similar to a study conducted by *Bessen et al. (2009)*; this involved educating clinicians in the use of the OAR and a specific ankle radiography request form, each designed to target barriers in the use of the OAR at an individual and institutional level. As expected, a significant change in practice was noted, with nurses demonstrating the greatest uptake in the OAR (*Bessen et al., 2009*). *Bessen et al. (2009)* also trialled the concept of "gatekeeping" among ED radiographers, who were empowered to reject request forms that demonstrated non-compliance with the OAR. These are important considerations in improving the compliance and hence accurate use of the OAR within the ED.

## CONCLUSIONS

In conclusion, although the OAR is internationally regarded as highly accurate clinical decision tool, its local uptake has been varied. The results of this audit demonstrate

moderate compliance and poor sensitivity of the rule. Despite limitations in the reporting of the diagnostic accuracy of the rule, this audit demonstrates reasonable evidence that individual and/or institutional-based change strategies are warranted to improve clinician compliance in the use of the OAR in the local tertiary care emergency department setting. Doing so will improve the implementation of the rule and reduce the frequency of radiography requests to optimise patient flow through the emergency department.

### Funding
The authors received no funding for this work.

### Competing Interests
The authors declare there are no competing interests.

### Author Contributions
- Yolanda E. Gomes conceived and designed the experiments, performed the experiments, analyzed the data, prepared figures and/or tables, authored or reviewed drafts of the paper, and approved the final draft.
- Minh Chau and Ryan S. Causby conceived and designed the experiments, analyzed the data, authored or reviewed drafts of the paper, and approved the final draft.
- Helen A. Banwell analyzed the data, authored or reviewed drafts of the paper, and approved the final draft.
- Josephine Davies performed the experiments, prepared figures and/or tables, and approved the final draft.

### Human Ethics
The following information was supplied relating to ethical approvals (i.e., approving body and any reference numbers):
University of South Australia Human Research Ethics Committee approved the study (Approval number: 202798).

### Ethics
The following information was supplied relating to ethical approvals (i.e., approving body and any reference numbers):
Southern Adelaide Local Health Network (SALHN) Office for Research—gave a waiver for this research. This is the overarching body for the Flinders Medical Centre Human Research Ethics Committee.

### Data Availability
Raw data is available in the Supplemental Files.

## Supplemental Information

Supplemental information for this article can be found online at http://dx.doi.org/10.7717/peerj.10152#supplemental-information.

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
