# Peer review of "Adequacy of clinical information in X-ray referrals for traumatic ankle injury with reference to the Ottawa Ankle Rules—a retrospective clinical audit"

_PeerJ, doi:10.7717/peerj.10152_

## Round 0.1 · original submission · Minor Revisions

Reviewer #2 wants minor revisions. Let's consider them in turn. First, the length of the Background section. From my point of view, all the paragraphs of this section are pretty brief as is, except for #6 (lines 80-93). Maybe you could shorten this one, and maybe even scour the whole section to eliminate needless words. Second, the requested discussion of the relationship between OAR and 'subjective examindor [maybe examiner?] perception'. Please consider this point, but keep remarks about the subject brief. Third, discuss 'the most important item of OAR'. If you can think of a useful remark to make, please add that, though I must say that the OAR seems to me (a non-clinician, naive to this whole business) very simple and brief altogether.

·

Basic reporting

The article presents the clinical problem, methods used and results of the study clearly. Professional English is used throughout the text. Authors provided abundant data regarding the background with corresponding references. Article is well-structured with appropriate use of figures and tables. Raw data was available for the review.
Results of the study are clearly presented and discussed in context of previous knowledge and the goal of the study.

Experimental design

Clinical problem of the article is well known in scientific community, but lack of data regarding local, Southern Australian region, can justify the research. It is important for both quality control on a local level as well as for comparison of results on a global scale to gather and publish information even regarding previously researched clinical problems. Experimental design is of a sufficient research integrity and reproducible.

Validity of the findings

The Authors admit that this is a novel research on a local level only, clearly underlying that there are some weaknesses of the study. Results seem statistically sound and are interesting to a medical professional. Discussion is well-written summarizing the results and possible impacts of the study.

Additional comments

The article offers a new insight to an old problem, offering possible reasons why Ottawa Ankle Rules application in Emergency department setting differs between medical professionals who refer patients for radiography. While not assessing the accuracy of the test itself, but rather in a context of different medical professions, this is still important information for scientific community.

·

Basic reporting

The manuscript is clear and well written. References, figure, and tables are adequate. Results, although not entirely new, are relevant.

Experimental design

The study is in accordance with the scope of the journal. Research question and hypothesis were adequately addressed, following all the required issues in terms of ethical standard. Methods were thoroughly described.

Validity of the findings

Although the findings are not entirely new, the results are interesting and worthy of disclosure. The manuscript results are adequately presented and compared with previous similar studies.

Additional comments

Dear authors, I appreciate the opportunity to review this interesting manuscript.
The topic is relevant in terms of cost-effectiveness regarding ankle trauma.
The article is well written and the methodology is adequate and thoroughly presented.
Ethical standards were also adequately addressed.
However, I have two minor suggestions aiming to improve the manuscript quality:
1-Background section is excessively long. Make it more straightforward and compare the previously published studies with your findings in the discussion section.
2- Discuss about the relationship between Ottawa Ankle Rules and subjective examinador perception to evaluate radiograph necessity following ankle sprain. Furthermore, some discussion should be presented in terms of the most important item of OAR to predict the presence of a foot/ankle fracture.

---

## Round 0.2 · accepted · Accept

As I had expected, you revised quickly and deftly. Well done.